# Terahertz Optics of Materials with Spatially Harmonically Distributed Refractive Index

**DOI:** 10.3390/ma13225208

**Published:** 2020-11-18

**Authors:** Dzmitry Bychanok, Gleb Gorokhov, Artyom Plyushch, Alfredo Ronca, Marino Lavorgna, Hesheng Xia, Patrizia Lamberti, Polina Kuzhir

**Affiliations:** 1Research Institute for Nuclear Problems Belarusian State University, Bobruiskaya str. 11, 220030 Minsk, Belarus; glebgorokhov@yandex.ru (G.G.); artyom.plyushch@ff.vu.lt (A.P.); polina.kuzhir@gmail.com (P.K.); 2Radioelectronics Department, Faculty of Radiophysics, Tomsk State University, 36 Lenin Prospekt, 634050 Tomsk, Russia; 3Faculty of Physics, Vilnius University, Sauletekio 3, LT-10222 Vilnius, Lithuania; 4Institute of Polymer, Composite and Biomaterials, National Research Council, Via G.PreViati, 1/E, 23900 Lecco, Italy; alfredo.ronca@cnr.it (A.R.); mlavorgn@unina.it (M.L.); xiahs@scu.edu.cn (H.X.); 5State Key Laboratory of Polymer Materials Engineering, Polymer Research Institute, Sichuan University, Chengdu 610065, China; 6Dept. of Information and Electrical Engineering and Applied Mathematics, University of Salerno, Via Giovanni Paolo II, 132-84084 Fisciano (SA), Italy; plamberti@unisa.it; 7Institute of Photonics, University of Eastern Finland, Yliopistokatu 7, FI-80101 Joensuu, Finland

**Keywords:** terahertz bandgap, graded index materials, harmonically distributed refractive index

## Abstract

The electromagnetic properties of structures with spatially periodic distributed graded refractive index were investigated in the terahertz frequency range. The band structure and electromagnetic response of material with harmonically distributed refractive index were calculated and analyzed. The analytical expressions for frequencies of the first and second bandgap are derived. 3D printed gyroid based architectures were proven to be harmonically graded refractive index structures with designed bandgaps in THz frequency ranges. The transmission coefficient of thermoplastic polyurethane-based samples were experimentally measured in the frequency range 100–500 GHz and compared with theoretical results. Due to losses in the real world produced samples, the predicted response is significantly dumped in the terahertz range and only traces of band gaps are experimentally observed. This funding paves the way toward a new generation of 3D printed THz components for gradient-index optics applications.

## 1. Introduction

The dielectric based periodic structures are widely investigated in the visible frequency range [1,2,3] and often may be considered as graded refractive index materials [4,5], being a practical realization of a gradient-index optics [6,7,8]. Graded refractive index materials can be found in nature (in particular, butterfly wings where colors appear due to a certain shape of chitin, gyroid) [4,9,10] and may be produced artificially [3,11].

Gyroid was first developed by NASA physicist Alan Schoen [12] as a result of his search for ultra-light and ultra mechanically robust material for space applications. A gyroid is a connected structure endlessly repeating in three dimensions that has the smallest possible surface. Passing through gyroid based butterfly wings, the light is refracted and we see such beautiful overflows and fantastic tones.

The gyroids are multifunctional structures–particularly in [13] they are proposed for designing bone scaffolds. The electromagnetic study of the gyroids structures of certain geometrical parameters will allow designing a new generation of components to manipulate light of different wavelength, including infrared and terahertz.

Nowadays the materials with a bandgap in the terahertz region have attracted a lot of attention of the scientific community [14,15,16]. The design, manufacturing, and tuning of such materials is a challenging problem for material science. Actually, the minimal surfaces were proposed in [17] as candidates for photonic bandgap materials. Recently, the mechanically tunable microwave properties of gyroid photonic crystal were investigated in [18].

In this communication, we suggest simple and reproducible road toward THz gradient-index optics by 3D printing of gyroid-based architectures as harmonically graded refractive index structures. We demonstrate both theoretically and experimentally the possibility to design the bandgap of 3D printed gyroid based structure in the submillimeter frequency range.

## 2. One-Dimensional Structures with Periodically Distributed Refractive Index

Let us consider material with harmonically distributed refractive index along z-axis (see Figure 1a):(1)$$n\left(z\right)=n_{0}+\Delta n\cos\left(2\pi z/\lambda \right),$$
where $n_{0}$ is the average value of the refractive index, Δn is the dispersion of refractive index, λ is the distribution period.

We start to study the features of propagation of electromagnetic waves along the z-axis in such material with the calculation of dispersion relation and band structure. The transformation of n(z) to ω(k) may be performed by using plane wave expansion method [19,20,21] (see details in Appendix A). The calculated band structure of considered material with parameters $n_{0}=1.26$, Δn=0.20 and λ=π/4 mm is presented in Figure 1b. One can see that the structure has at least two bandgaps near 150 and 300 GHz. The detailed inspection of Figure 1b shows that exist also very narrow (the width less than 1 GHz) the third bandgap near 450 GHz.

The electromagnetic response (i.e., relative amplitudes of reflected $S_{11}$ and transmitted $S_{21}$ plane waves normally scattered on the sample of finite thickness τ) of material with spatial refractive index distribution may be calculated using matrix method [22,23] (see Appendix B). The results of S-parameters calculations for τ=3π mm are presented in Figure 1c. The material becomes to be not transparent and electromagnetic waves are reflecting from it near 140–163 GHz. The second and third reflection and transmission peaks also exist at the frequencies near 300 GHz and 450 GHz. The most pronounced is the first peak, the second one has lower amplitude and is more narrow, the amplitude of the third peak (and all next ones) becomes almost comparable with the interference oscillations related to the finite size of a sample τ.

The physical origin of the observable band structure is based on interference of reflected from investigated sample waves. The position of the band gaps is generally determined by the averaged value of the refractive index and the distance between the maxima of the periodic function *n*. Partial waves reflected from regions separated by distance λ are added to each other and form a signal reflected from the sample. When the phase difference between them reaches 2π, the first bandgap appears. In this case, the effective wavelength is 2λ=π/2 mm, which corresponds to frequencies of about 150 GHz. Waves reflected from neighboring regions, in this case, are in phase and amplify each other. This is the origin of why material becomes opaque for frequencies within the bandgap.

The general feature of presented in Figure 1b,c dispersion relation and electromagnetic response is two pronounced bandgaps near 140–163 GHz and 300 GHz. Important to note that the gaps are equidistant and the second bandgap (also the third and all next ones) is significantly less than the first. This type of band structure is a specific feature of structure with refractive index *n* distribution by Equation (Equation 1) in contrast to, for example, a photonic crystal formed by a one-dimensional array of air slabs penetrating a dielectric background [19].

In the sections below we will examine how the position and width of the lowest two gaps depend on the average value of the refractive index, its dispersion and distribution period.

### 2.1. Dependence of Gap on *n*, Δn and λ

#### 2.1.1. Numerical Results

According to numerical calculations the first bandgap in Figure 1 is about 23.8 GHz the second is 5.6 GHz for $n_{0}=1.26$, ΔN=0.20 and λ=π/4 mm. To estimate how the electromagnetic response depends on parameters in Equation (Equation 1) we first of all performed the simplest numerical experiment and calculated the spectra of $S_{11}$ and $S_{21}$ varying *n*, Δn and λ. In general, the structure of spectra remains the same as the Figure 1c, but the position and width of the bandgaps are changed.

Particularly, by increasing $n_{0}$ in two times, the middle frequencies of bandgaps are shifting down twice, and the bandgap width suppresses approximately 4 times. For $n_{0}=2.52$ the first bandgap is about 6.0 GHz and locating near 75 GHz. The second bandgap is about 0.7 GHz and locating near 150 GHz.

The decrease of Δn in two times remains the middle frequency of bandgaps the same and width of the first gap decreases by about two times. For Δn=0.1 the first bandgap is about 12 GHz near 150 GHz. The second bandgap is about 1.4 GHz and locating near 300 GHz.

The decrease λ in two times shifts the frequency up twice and the width of both bandgaps is increased by about two times. For λ=π/8 the first bandgap is about 47.7 GHz and locating near 300 GHz. The second bandgap is about 11.3 GHz and locating near 600 GHz.

Analyzing the numerical experiments results, one may conclude that the width of the first gap is $\Delta \nu_{1}∼\Delta n/\left(n_{0}^{2}\lambda \right)$, for the second gap $\Delta \nu_{2}∼\Delta n^{2}/\lambda$. These equations are intuitive relations obtained from numerical experiments based on the matrix method [22,23]. Their exact values will be derived in the next section using the wave expansion method.

#### 2.1.2. Analytical Results

The method of band structure calculation described in Appendix A is valid for arbitrary periodic distribution of refractive index along the z-axis. Throughout the article we use SI units and assumed a exp[iωt−ikz] harmonic time convention. In the case of harmonic distribution Equation (Equation 1), the coefficients $k_{m}^{\varepsilon }$ in the Fourier expansion (see Equation (Equation 10)) are forming the series with very fast decreasing terms (in considered in Figure 1 example, they are 0.654, −0.104, 0.012, −0.0013 for *n* = 0, ±1, ±2, ±3, correspondingly). The general contribution may be considered by using only first terms *n* = 0, ±1. In this case, there exists a relatively simple equation for dispersion relation ω(k). To calculate ω(k) it is necessary to solve the following equation (see details of derivations in Appendix A): (2)$$\det\left(\begin{array}{ccc} n_{0}^{2}+\frac{1}{2}\Delta n^{2}-\frac{c^{2}}{\omega^{2}}{\left(-\frac{2\pi }{\lambda }+k\right)}^{2} & n_{0}\Delta n & \frac{1}{4}\Delta n^{2}\\ n_{0}\Delta n & n^{2}+\frac{1}{2}\Delta n^{2}-\frac{c^{2}}{\omega^{2}}k^{2} & n_{0}\Delta n\\ \frac{1}{4}\Delta n^{2} & n_{0}\Delta n & n^{2}+\frac{1}{2}\Delta n^{2}-\frac{c^{2}}{\omega^{2}}{\left(\frac{2\pi }{\lambda }+k\right)}^{2} \end{array}\right)=0,$$
where *n*, Δn and λ are parameters from Equation (Equation 1), ω=2πν is angular frequency, *c* is the speed of light.

The roots of Equation (Equation 2) with k=π/λ gives frequencies for the first bandgap. In case of the second bandgap (substituting k=0 in Equation (Equation 2)) it is possible to simplify significantly the Equation (Equation 2) and obtain simple analytical expression for the second gap:(3)$$\Delta \nu_{2}=\nu_{3}-\nu_{2}=\frac{2c\sqrt{2n_{0}^{2}+\Delta n^{2}}}{\lambda \sqrt{8n_{0}^{4}-6\Delta n^{2}n_{0}^{2}+\Delta n^{4}}}-\frac{2c}{\lambda \sqrt{4n_{0}^{2}+\Delta n^{2}}}.$$

Equations (Equation 2) and (Equation 3) provide very good correspondence with the numerical calculations Via matrix method and may be applied for quick estimation of the frequencies $\nu_{1}$, $\nu_{2}$, $\nu_{3}$ and corresponding bandgaps $\Delta \nu_{1}$, $\Delta \nu_{2}$.

The mentioned above band structures of materials with a harmonically distributed refractive index may be useful for terahertz applications. Particularly, the realization of structures with a distributed refractive index may be obtained using 3D-printing technology. In the sections below, we propose to use gyroid-based 3D-printed structures as an example of spatially distributed refractive index material. The specialty in gyroids is their unique form, which provide periodic cosine-like distribution of effective refractive index in this material. Exactly this feature provides a simple way to form a periodic structure with predictable, analytically solvable, and equidistant band structure for a wide range of terahertz applications.

## 3. Gyroid Stuctures Production

The gyroid is an infinitely connected triply periodic minimal surface described in the Cartesian coordinate system by equation [24]:(4)sin(x)cos(y)+sin(y)cos(z)+sin(z)cos(x)=0.

To obtain volume structure from surface described by Equation (Equation 4) the following inequality should be used:(5)sin(αx)cos(αy)+sin(αy)cos(αz)+sin(αz)cos(αx)>C,
where *C* is parameter related to the average porosity of the structure [25] and α is the scaling factor. Below we will use the scaling factor α=2 mm${}^{-1}$. The typical gyroid-based structure with porosity 60% related to $C_{1}=0.27$ is presented in Figure 2a. The cross-sections perpendicular to z-axis are presented in Figure 2b. They are forming periodic patterns of air voids and material areas depending on depth αz.

The presented in Figure 2 structure may be produced by 3D-printing technology, which offers unprecedented opportunities for production complex periodic structures with advanced mechanical, thermal and electromagnetic properties [26,27,28,29]. Particularly, here the selective laser sintering (SLS) technology was used. The details of the preparation were discussed in [25]. Particularly the micrometer-sized thermoplastic polyurethane (TPU) particles were sintered using laser beam layer-by-layer within the area satisfying Equation (Equation 5). This strategy allows forming from TPU powder the gyroid structure with porosity about 60%. The lateral dimension of 3 period thick printed samples was about 3π≃9.5 mm.

## 4. Scattering of the Electromagnetic Wave

The scattering of electromagnetic waves in millimeter and submillimeter frequency ranges on studied gyroid structures may be considered within the longwave approximation. In practice, this approach is well applicable for the normal scattering of a plane wave on not very thick periodic arrays infinite in directions perpendicular to the initial wave vector [30,31].

To apply this approach and calculate the transmitted and reflected signals along to the initial wave vector direction it is necessary to consider the average density variation within the unit cell of the structure. Let us consider the cross-sections of presented in Figure 2a structure in the plane perpendicular to z-axis (see Figure 2b). The dark areas are related to the regions filled with material, bright ones are air. Depending on position *z* there is a variation of air and material fractions inside the sample. Averaging the refractive index according to the air and material fractions it is possible to obtain its effective distribution of $n_{eff}$ along the z-axis. The similar averaging procedure was performed in [13] for effective density calculations. In this case $n_{eff}\left(z\right)$ may be calculated as [4]:(6)$$n_{eff}=n_{bulk}\left(1-S\left(z\right)/4\pi^{2}\right)+n_{air}S\left(z\right)/4\pi^{2},$$
where S(z) is the air surface in cross-section plane perpendicular to z-axis within the unit cell αx∈[−π,π], αy∈[−π,π], $n_{bulk}$ is the refractive index of bulk material, the term $4\pi^{2}$ is the total surface of the unit cell, $n_{air}=1$.

In the case of gyroid S(z) may be easily numerically calculated by considering one unit cell by Quasi Monte-Carlo method [32]. The results of calculation are presented in Figure 3a.

From this figure, we can see that average porosity of considered structure within the unit cell (see inset of Figure 3a) is varied near 60% [25]. Nevertheless, the penetration inside the sample along z-axis shows that there are areas within the structure which are more or less dense than the average value. The variation of density is relatively small and its maximum deViation is about 5%.

The calculated distribution of refractive index using Equation (Equation 6) is presented in Figure 3b, where we used the typical for TPU value of refractive index $n_{bulk}=1.64-i0.06$. This distribution will be used below to calculate transmitted and reflected signals along z-axis of three periods thick layer (see Figure 3c) formed by gyroid infinitely translated along x- and y-axes (total thickness of structure was 3π mm).

## 5. Terahertz Measurements in Free Space

The time-domain spectrometer “T-Spec” produced by EKSPLA was used for measurements of the electromagnetic response of printed samples in the 100–500 GHz range. A pumping 1050±40 nm wavelength laser with 40 mW averaged output power was used to excite a photoconductor antenna and produced THz radiation pulse with an aperture about 8 mm, normally scattered on the investigated sample. The experiments were performed in transmission geometry. The Fourier transform of the measured waveform transmitted through the sample THz pulse gives the frequency dependence of the complex transmission coefficient of the investigated material. The measurement procedure was in details described in our recent works related to terahertz properties of nanocarbon based structures [33,34]. The results of measurements of the transmitted through the sample signal in the logarithmic scale $S_{21log}=20\mathrm{lg}\left(S_{21lin}\right)$ are presented in Figure 4 (bold blue line).

The losses in the real TPU have significantly dumped the interference and transmission is essentially decreased with frequency. Nevertheless, Figure 4 shows that traces of the band structure are observed near the frequencies of 150, 300 and 450 GHz. The bend in experimental and theoretical spectra is most probably due to the difference of real refractive index distribution in gyroid (Figure 3) to harmonic Equation (Equation 1). At higher frequencies about 500 GHz the wavelength becomes less than 0.5 mm and measured spectra differs significantly from predicted within the longwave approximation results.

## 6. Conclusions

The obtained results show that structures with a harmonically distributed refractive index might be utilized as materials with controlled bandgap in the terahertz frequency range. In particular case of material with $n_{0}=1.26$, Δn=0.20 and λ=π/4 mm the bandgaps are locating near frequencies 150 GHz, 300 and 450 GHz. The analytical model demonstrates that the width of the bandgaps is generally defined by parameter Δn=0.20. The bandgaps positions may be shifted to a higher frequency range by decreasing of period λ or by applying structures with higher porosity (i.e., higher air fraction and correspondingly lower $n_{eff}$).

The gyroid based architectures were proposed as a practical realization of harmonically distributed refractive index structures. The variation of average density within the unit cell provides a cosine-like distribution of refractive index in such structures within the longwave approximation. The presence of three equidistant bandgaps was experimentally proved in 3D printed thermoplastic polyurethane-based gyroid.

It is important to note that without losses the expected spectra of produced sample are very similar to that presented in Figure 1c. In the terahertz region, the losses in the polymer matrix become significant for samples with a thickness of more than several millimeters. For thick samples, the transmission substantially decreases with frequency. This tendency is clearly seen in Figure 4. Due to dumping, the partial waves deeply penetrating the material have reduced impact on the overall electromagnetic response. The interference oscillations are smoothing and the band structure becomes less visible. Actually, in the experiment (Figure 4) we observed only “traces” of the band structure theoretically predicted. Nevertheless, the positions of peaks correspond to the expected frequencies.

The difference between experiment and modeling may be explained by the two reasons. First, the refractive index of TPU is frequency-dependent n=n(ν) in the considered 100–500 GHz frequency range. The systems with n(ν) are of great practical interest but require additional investigation. Another reason is the one-dimensionality of the considered model. For a more accurate wave scattering calculation, complex numerical simulations based on the Huygens-Fresnel principle are required.

To reduce the losses’ impact it is necessary to decrease the thickness of the sample with the same number of periods. In this case, the band structure will be much more pronounced. Therefore the need to reduce the period λ is clear. However, most printers available today do not allow producing high quality samples in gyroid geometry with a period of less than 0.5 mm. The obtained results prove the concept of THz optics of graded refractive index materials and demonstrate the importance of miniaturization for THz applications. They open the route stimulating the development of new 3D printing technologies with higher resolution and smaller characteristic size. For further applications it is also necessary to develop 3D-printable polymer materials with low losses in the THz range. We hope that the results obtained in this work will be soon implemented on thinner samples and will stimulate the development of terahertz optics of graded refractive index materials. 

## Figures and Tables

**Figure 1 materials-13-05208-f001:**
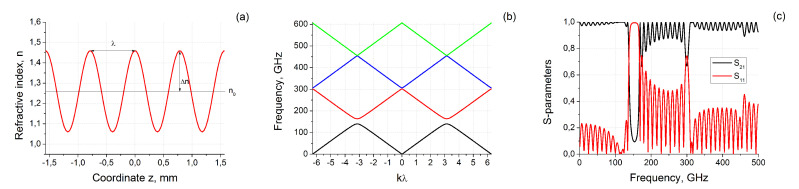
(**a**) The refractive index distribution along z-axis; (**b**) The band structure of considered material with parameters $n_{0}=1.26$, Δn=0.20 and λ=π/4 mm; (**c**) relative amplitudes of reflected ($S_{11}$) and transmitted ($S_{21}$) signals of piece of material with thickness τ=3π≃9.5 mm.

**Figure 2 materials-13-05208-f002:**
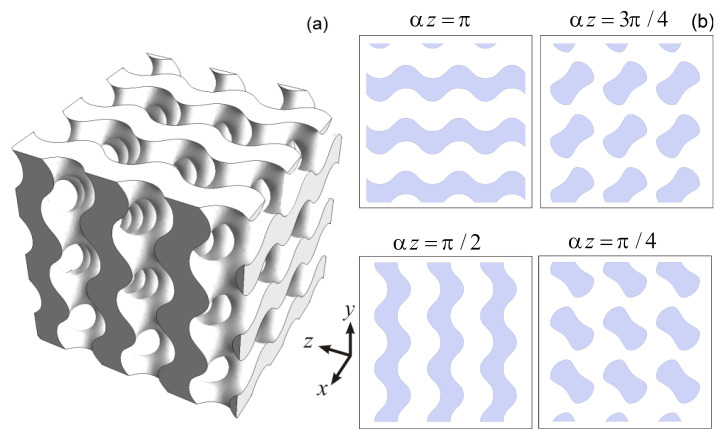
(**a**) The gyroid-based structure with porosity 60%; (**b**) The cross-sections of gyroid structure with porosity 60% in the plane perpendicular to z-axis (gray regions are related to material, white—to the air).

**Figure 3 materials-13-05208-f003:**
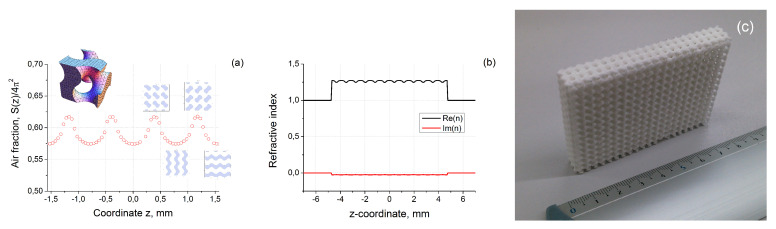
(**a**) Relative air fraction distribution in gyroid along the z-axis (Inset: single unit cell of the structure and characteristic cross-sections of periodic material); (**b**) calculated distribution of refractive index about z-axis; (**c**) printed sample.

**Figure 4 materials-13-05208-f004:**
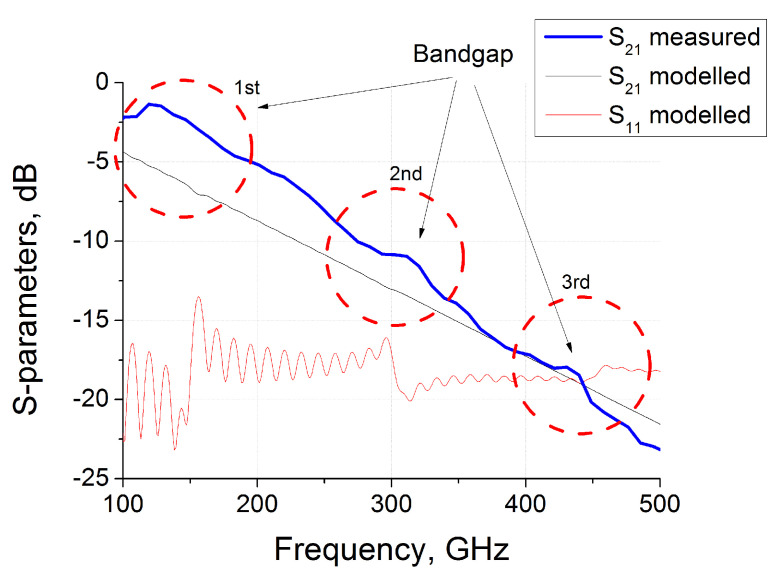
Experimentally measured and numerically calculated *S*-parameters of 9.5mm-thick gyroid samples in the free space.

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
