# Peer review of "Terahertz Optics of Materials with Spatially Harmonically Distributed Refractive Index"

_materials, 2020, doi:10.3390/ma13225208_

Round 1

Reviewer 1 Report

See attachment

Reviewer 2 Report

The authors tried to develop some kind of theoretical relation between n_0, delta-n, and lambda and index of refraction distribution which seems to be the main conclusion.
Even tough the result is not far from the one expected based on our intuitive picture, rigid calculation could have some meaning for the future applications, if it is well proved.

In the later part, authors synthesized the real-world sample with gyroid and performed THz measurement.
The result of photonic band gaps from calculation and experiment are roughly matching which might verify one of the theoretical result.
However, this experiment do not support the conclusion of the manuscript.
Authors need to do the same experiment with other gyroids with different configuration to prove the relation between n_0, delta-n, and lambda and phonic band gaps.
Otherwise, the experimental work only proves that the gyroid can be use as a photonics crystal which is already demonstrated many times before.
I also do not find any specialty in gyroids in this context, since similar photonics band gaps can be formed in many other photonics crystals, such as superlattices or split ring resonators.

Reviewer 3 Report

In their manuscript Bychanok and coauthors report 3D printed gyroid structures as graded-index strructures with tunable bandgaps in the THz range. Overall, their results are robust and in good agreement with theory. I therefore recommend the manuscript for publication, provided that a few comments are addressed:

How do the authors find the equations in line 80? Do they just follow from their empirical model or is there a deeper theoretical reason? If not a would suggest to put the values they describe in section 2.1.1 ona plot and perform a fit.

Is the output power of their pulsed laser (40 mW) the average or peak one?

In general, the English level has to be improved throughout the whole manuscript, as many articles are missing, the use of many prepositions is a bit weird, and the position of verbs, nouns, and adverbs in the sentences is wrong. Here below are a few typos I found:

  • Line 30: repetition of "properties"
  • Line 60: background
  • Lines 70,71: rephrase, they are a bit confusing
  • Line 120: choose one between thick and dense
  • Line 124: too many "consider"
  • Line 139: please rephrase, it's a bit confusing

Round 2

Reviewer 2 Report

The authors did improve the manuscript at some places, the theoretical considerations are sufficient, the experimental part, however, remain below average. Fig. 4 does not really support any agreement between theory and experiment. I do see some changes at frequencies expected, but the calculated S21 looks rather different to the experimental one. This discrepancy is not discussed. In general, the discussion at this point should be extended, as the authors put the transmission measurements in the title of their paper.

I do not support publication of this manuscript, albeit I do not want to object.

Minor point:

in line 56 the authors talk about material waves when considering electromagnetic waves.
